# Microbiome of Sri Lankan Coral Reefs: An Indian Ocean Island Subjected to a Gradient of Natural and Anthropogenic Impacts

Mohamed F. M. Fairoz [1,*], Kevin T. Green [2], Kuwaja N. M. Sajith [1], Weerathunga A. S. Chamika [1], Amarasingha M. K. N. Kularathna [1], Saichetana Macherla [2], Douglas S. Naliboff [2], Ana Cobián-Güemes [2], Linda Wegley-Kelly [2] and Forest Rohwer [2]

[1] Department of Fisheries and Marine Sciences, Ocean University of Sri Lanka, Mahawela Road, Tangalle 82200, Sri Lanka
[2] Department of Biology and Viral Information Institute, San Diego State University, San Diego, CA 92182-4614, USA
* Correspondence: fairoz.mfm@gmail.com

**Abstract:** Coral reefs around Sri Lanka have coexisted with human communities for thousands of years and are a continual source of food, economic productivity, and tourism. Although these reef systems sustain nearby populations, little is known about the presence or functional role of microbial communities on reefs dominated by hard corals or fleshy algae. Coral reef benthos cover was recorded, and reef-associated water samples were collected, sequenced and analyzed from seven coral reefs around Sri Lanka. Microbial metagenomes were analyzed to reveal both the taxonomic and metabolic makeup of the microbial communities present at each site. A metagenomic analysis of microbial phyla showed that Alphaproteobacteria and Gammaproteobacteria were most abundant, constituting up to 79.4% of microbial communities. At the order level, Rhodobacterales dominated the microbial communities across all sites, with the exception of the Paraviwella coral reef, where the order Alteromonadales dominated. A Principal Component Analysis (PCA) was performed using metagenomic sequence data to find the possible trends of interactions and drivers of taxonomic and metabolic community structure. This study is the first microbial metagenome dataset of coral reef associated water from the Indian Ocean continental island, Sri Lanka. These data further confirm the need for a comprehensive study of reefs in Sri Lanka aimed at elucidating the processes involved in microbial energy utilization.

**Keywords:** Indian Ocean; Sri Lanka; microbiome; human impacts; coral cover; bacterial taxonomy

## 1. Introduction

The Indian subcontinent is a physiographical region in South Asia situated on the Indian Plate, projecting southwards into the Indian Ocean from the Himalayas. Sri Lanka is a large continental island off the southeast coast of the Indian subcontinent located 35 km from India at its most northwestern end, 435 km long, 225 km wide and has a total area of 66,580 km$^2$ inhibited by about 21 million people [1]. The coastal regions are make up 67 administrative districts [2] and contain about 35% of the total human population of Sri Lanka.

Coral reefs are present in varying conditions, and coral diversity has previously been reported for Sri Lankan reefs with total of 90 species of stony corals belonging to 39 genera, 27 of which included 70 homotypic species [3–6]. There were 36 new coral species confirmed by DNA bar coding for Northern Sri Lanka [7].

Reefs in Sri Lanka have been increasingly exposed to deleterious natural events and anthropogenic activities in the recent decades. These events have resulted in reef health decline, and Sri Lankan reefs have been recently classified as some of the most exploited and degraded in the region. While some reefs face decline and extinction, others in the

country are considered some of the most untouched in the Indian Ocean [8,9]. Several coral bleaching events in 1998 and subsequent bleachings in 2000 and 2015, along with the physical damages to reefs from Tsunami 2004 [10], are responsible for the decrease the live coral cover in the region [11]. Reefs in the northeastern region were less affected by tsunami and recovered from bleaching events due to the localized upwelling [12] and the offshore locations of reefs.

Direct and indirect human activities have contributed to reef degradation [13]. This was noted by increased algal cover following that the supply of pollutants in freshwater runoff from rivers and associated turbidity from monsoonal changes near the coral reefs in Sri Lanka. Several rivers discharge heavy sediment loads that originated from unplanned land reclamation with polluted materials [14,15]; this process is exasperated during the rainy seasons [16,17].

Previous works to better understand the current reef condition and implement improved management practices around Sri Lanka found the reefs in Southern Sri Lanka to be in poor health due to human impact [14–16]. Reefs were vulnerable [17] due to activities such as coral mining, destructive fishing, ornamental reef fish collection [16], tourism and pollution.

The reefs along the northwestern and eastern coasts are the least impacted due to previous internal conflicts that displaced up to 73,700 people to the south [18] and recent increased governmental patrol of the coastal regions. In contrast, increased population and commercial development along the southern coast have had deleterious effects on reef health [9].

Research performed previously in Sri Lanka has focused on the benthic cover with respect to water chemistry and microbial abundance. These data were included in a global study representing the Indian Ocean [19]. This study showed that the reef degradation was not directly affected by increased nutrients (nitrogen and phosphorus) but that dissolved organic carbon (DOC) will cause coral mortality as a result of increased microbial activity and subsequent hypoxia [20]. The benthic (coral cover and macroalgae) cover, microbial abundance and water chemistry data contributed from Sri Lankan coral reefs were incorporated to explain the global microbialization of coral reefs with the DDAM (DOC, disease, algae and microbes) model [19]. The analysis showed a strong correlation between bacterial biomass, DOC and nutrients to macroalgae and corals. Further, the reef locations in the north and south were clustered in relation to human population densities, confirming that the regional differences of coral reefs health. Therefore, the present metagenomics data representing the coral reefs from the northeast and south could explain the gradient of the reef status. An extensive study performed in Northern Line Islands (Pacific Ocean) proved that that the connectivity to microbial ecology of reefs with the remoteness and presence of human populations is interlinked and explained the status of reef health [21–24].

Our goal was to initiate coral reef microbiome studies on Sri Lankan coral reefs using metagenomic sequencing and a bioinformatic approach to elucidate the microbial taxonomic and functional composition. The reefs were monitored by sampling the water column at seven reef sites in two main regions (northeast and south) along coastal Sri Lanka. The northeast reef sites represented Kuchchaweli (KU), Coral Island (CI), Pigeon Island (PI) and Small Island (SI) reefs. The south was represented by Paraviwella (PV), Polhena (POL) and Weligama (WEL) reefs. This paper is the first metagenomic sequencing data reporting the microbial communities present at coral reefs located in the northwest and south of Sri Lanka associated with a wide spectrum of coral covers across Sri Lanka under the human population density proximity to reefs. This study provides an opportunity to understand the microscale reef processes and microbial energy dynamics to recognize prime reef areas to be sustainably managed and initiate conservation priorities.

## 2. Materials and Methods

### 2.1. Reef Locations Selected for Sample Collection

Northeast samples were collected in June of 2015, and southern samples were collected between February and March of 2016 (Figure 1A). Geographic coordinates and a month of sampling for each of the seven sites are provided in Tables 1 and 2, respectively. All reef sites in this study were previously included in monitoring studies [4–6,11,13].

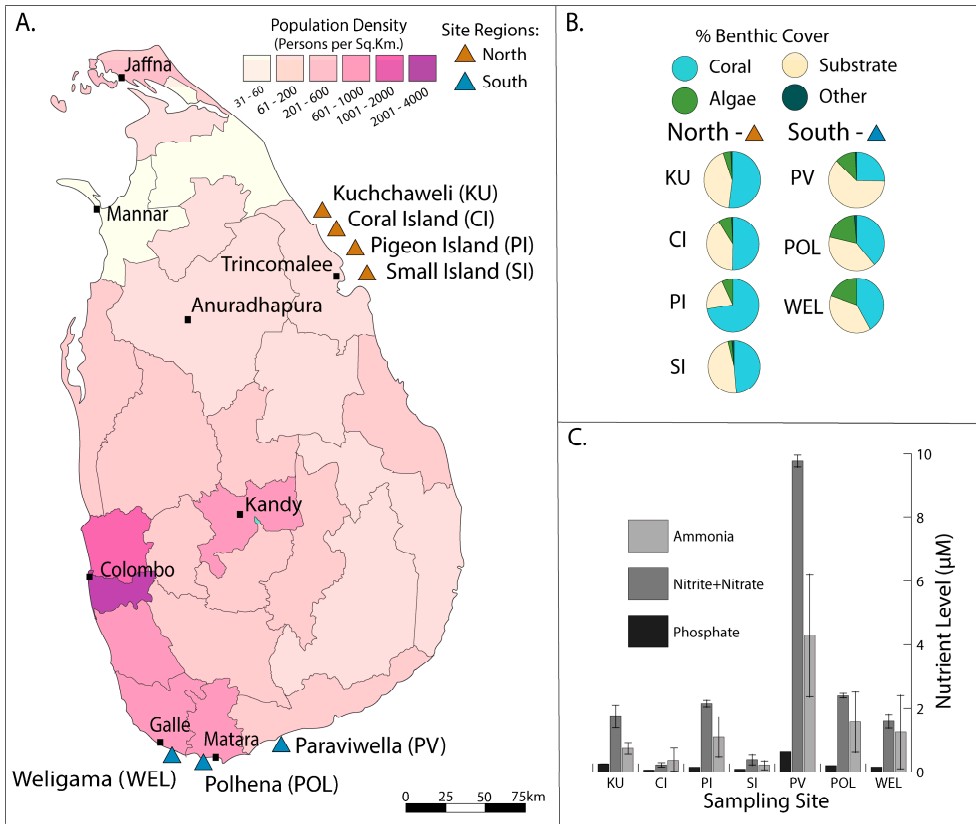

**Figure 1.** Overview of Sri Lanka administrative boundaries, status of benthic cover and water chemistry of sampling sites located in the northeastern and southern administrative districts. (**A**) Map of Sri Lanka highlighting major cities, including the district boundaries shown in lines, which were used for administrative purposes. Districts are the second level administrative divisions of Sri Lanka, preceded by provinces. Sri Lanka has 25 districts organized into 9 provinces. Coral reef sites have brown triangles for the northeast and blue triangles for the south. The population density by districts shows that the northeast coral reefs studies fall inside the Trincomalee District belonging to the eastern province, the 2nd lowest population density range representing 61–200 persons per square kilometer. Reef sites in the south are represented by the Hambantota, Matara with a southern province population density 601–1000 persons per square kilometer. (**B**) Percent cover of benthic communities at each reef sample site in Sri Lanka. Percent live coral cover around Sri Lanka ranged from 16.5% at Paraviwella reef (PV) to 69.8% at Pigeon Island (PI); the mean across all sample sites (mean ± SD) was 43.8 ± 17.2%. The percent algae cover ranged from 6.4% at Small Island (SI) reef to 14.9% at Weligama reef (WEL). The algal cover was lower in the northeast (ranging from 6.4 to 10.46%, mean ± SD 8.5 ± 1.6%) than in the south (ranging from 10.1 to 14.9%, mean ± SD 12.6 ± 2.4%). Weligama reef (WEL) and Polhena reef (POL) in the south belong to Matara District. Paraviwella reef site (PV) in the south belongs to Hamabnatota District. (**C**) Inorganic nutrients measured at each site when samples for all reef sites were studied. The PV reef is exposed to untreated sewage outflow and is in close proximity to a fishing harbor. POL and WEL reefs were found closer to estuaries receiving catchment from the Nilwala and Polwaththe Rivers, respectively.

**Table 1.** GPS coordinates of coral reef sampling sites around Sri Lanka included in the present study to determine the percent cover of benthic communities, water chemistry and microbial abundance and microbial metagenomics. Kuchchaweli (KU), Coral Island (CI), Pigeon Island (PI) and Small Island (SI) belong to Trincomalee District. Weligama reef (WEL) and Polhena reef (POL) in the south belong to Matara District. Paraviwella reef site (PV) in the south belongs to Hamabnatota District.

| Site | Coordinate North | Coordinate East |
|---|---|---|
| Kutchchaweli (KU) Trincomalee District | 08°49′45.48″ N | 81°6′15.958″ E |
| Coral Island (CI) Trincomalee District | 08°44′31.931″ N | 81°10′45.073″ E |
| Pigeon Island (PI) Trincomalee, District | 08°43′18.764″ N | 81°12′16.509″ E |
| Small Island (SI) Trincomalee, District | 08°42′48.232″ N | 81°12′3.06″ E |
| Paraviwella (PV) Hambantota District | 06° 1′ 17.503″ N | 80°48′8.191″ E |
| Polhena (POL) Matara District | 05°56′5.65″ N | 80°31′32.305″ E |
| Weligama (WEL) Matara District | 05°34′24.996″ N | 80°15′10.548″ E |

**Table 2.** Mean percentage cover of live corals, macroalgae and other habitat components recorded for the coral reef sites within the northeast and south of Sri Lanka from June 2015 to March 2016. Dead coral and soft coral were categorized as substrates. All other abiotic components were included as substrate rock, coral rubble, silt and sand. Kuchchaweli (KU), Coral Island (CI), Pigeon Island (PI) and Small Island (SI) belong to Trincomalee District. Weligama reef (WEL) and Polhena reef (POL) in the south belong to Matara District. Paraviwella reef site (PV) in the south belongs to Hamabnatota District.

| Region | Site | Live Coral | Macroalgae | Substrate ** | Other * |
|---|---|---|---|---|---|
| Northeast | KU | 52.6 ± 7.4 | 10.4 ± 1.9 | 31.5 ± 2.2 | 5.4 ± 2.15 |
| | CI | 50.3 ± 4.1 | 8.4 ± 4.7 | 36.0 ± 1.5 | 5.2 ± 1.65 |
| | PI | 69.7 ± 6.9 | 8.9 ± 3.0 | 17.9 ± 3.9 | 3.3 ± 4.0 |
| | SI | 49.7 ± 8.4 | 6.4 ± 0.4 | 38.5 ± 0.8 | 5.3 ± 2.6 |
| South | PV | 16.4 ± 10.1 | 10.1 ± 3.3 | 68.3 ± 5.4 | 5.1 ± 2.2 |
| | POL | 31.4 ± 12.8 | 12.8 ± 2.9 | 49.4 ± 1.0 | 6.2 ± 2.5 |
| | WEL | 36.2 ± 14.9 | 14.9 ± 3.4 | 46.5 ± 3.5 | 2.2 ± 4.7 |

**—dead coral and soft coral, and *—rock, coral rubble, silt and sand.

*2.2. Water Sample Collection and Processing*

Seawater samples were collected using four diver-deployable 2 L Niskin bottles that were pre-washed and capped until sampling. At each reef location, 8 L of seawater were collected from ~1 m above the reef surface at a water depth of ~3 m [22]. Prior to sampling; all equipment (except the GF/F filters) was submerged for 24 h in a 5% hydrochloric acid (HCL) bath to remove possible DOC contaminants [21]. Care was taken to be upstream from other divers and boats to avoid contamination. At the time of sample collection water temperature (at all south sites and PI in the north) and salinity (at all sites) were measured using a HOBO temperature logger and optical refractometer, respectively.

Water samples were processed within 3 h of collection to produce subsamples for direct counts of microbial and Archaea; nutrient analysis (ammonia, nitrate + nitrite and phosphate) and metagenomic sequencing. Briefly, four Hatay Niskin bottles [20,22] (a total of 8 L) were connected to acid washed silicone tubing and polycarbonate fixtures, and 200 mL of sampled seawater was flushed through the system prior to sample processing. This filtration system minimizes external contamination and facilitates efficient filtration of large volumes of water (see videos in [22]).

After collection of samples for nutrient and direct microbial count analyses, 1.5 L of sample water (375 mL from each of four Niskins) was passed through a 0.22 μm Sterivex filter to retrieve the microbial communities present. Excess water was flushed out of the filter using a 60 mL syringe prior to filter transport and storage at −20 °C. The 1.5 L sample volume has been sufficient for previous metagenomic analyses [19,21–24].

### 2.3. Population Density of Sri Lanka

Population statistics were collected from the official 2012 Sri Lankan census performed by the Department of Census and Statistics, Ministry of Finance and Planning [25]. This is the available census data to date approved by the authorities. The population density of each district is reported in persons per square kilometer ($km^2$).

### 2.4. Percent Benthic Cover Determination

Benthic cover at each site was determined with photo-quadrats as previously described [19]. At each coral reef, between March 2014 and October 2017, two 25 m transects were sampled and ten photo-quadrats were randomly selected for analysis. Images were taken with a digital camera 1 m above the reef surface and were later analyzed using Photogrid 1.0 software. Each image was analyzed by creating 100 stratified random points on the image; all benthic organisms captured and marked by a point on the image were identified using photographic references [26–28]. The percentage cover for each of four broad categories was reported: coral (live coral, Hexacorallia and Octocorallia); algae (nutrient indicator algae: fleshy algae green, red, orange or brown); substrate (recently killed coral, rock, silt/clay and sand) and other (sponges). The percent cover was calculated by dividing the number of points that were assigned to the respective functional group by the total number of points counted for each photo-quadrat. To determine the average percentage cover per site, the mean abundance for each category was determined across the 20 photo-quadrats.

### 2.5. Microbial and Archaea Abundances

The abundances of microorganisms (bacteria and virus-like particles [VLP's]) in each sample was determined by direct counts using a SYBR gold staining procedure and epifluorescence microscopy as previously described [22]. A portion (5 mL) of each sample was preserved in 16% paraformaldehyde during subsample preparation in the field. Preserved and stained sample (500 μL) was passed through a 0.02 μm Anodisc (Whatman, Maidstone, UK) and microbial particles enumerated using epifluorescence microscopy.

### 2.6. Environmental Parameters and Nutrient Concentrations

Temperature and salinity were measured using an in situ temperature logger (HOBO) and an optical refractometer, respectively. Inorganic nutrient (nitrate + nitrite, ammonia and phosphate) concentrations were measured using a QuikChem 8000 flow injection analyzer (Lachat Instruments) at the Marine Science Institute Analytical Laboratory (University of California, Santa Barbara, CA, USA).

### 2.7. DNA Extraction

DNA extraction and sequencing was performed as previously described [23]. DNA was extracted from the organisms collected on a 0.22 μm Sterivex filter using the Nucleospin Tissue Kit (Macherey-Nagel; Dueren, Germany) following the manufacturer's protocol.

Briefly, 0.22 μm Sertivex filters were thawed, and excess water inside the filter was flushed via a 10 mL Luer-Lok syringe. T1 lysis buffer (410 μL) with 20 mg mL$^{-1}$ Proteinase K was added, and both ends were closed with a Luer Thread Style Plug (Nordson MEDICAL; Marlborough, MA, USA) and covered in parafilm. The closed filters were placed into a 55 °C oven on a rotating spit to incubate overnight. Following overnight incubation, 400 μL of Buffer B3 was added into each filter, then placed back into the rotating oven to incubate at 70 °C for 30 min. The lysate produced by incubation was recovered from the 0.22 μm Sterivex filter using a 3 mL Luer-Lok syringe and transferred into a clean, sterile 1.5 mL microcentrifuge tube. To each tube containing the lysate, 420 μL of 100% ethanol were added, and total DNA was recovered via silica filtration as described in the manufacturer's protocol. The Qubit High-Sensitivity dsDNA kit (Life Technologies; New York, NY, USA) and NanoDrop (Thermo Scientific; Pleasanton, CA, USA) were used to determine the DNA concentration and purity, respectively.

## 2.8. DNA Sequencing

Sample libraries were prepared using the Nextera XT DNA library Prep Kit (Illumina; San Diego, CA, USA), and the manufacturer's protocol was followed to prepare the library for sequencing. Purified sample DNA was diluted to 0.2 ng μL$^{-1}$ from the original concentration, and a total of 1 ng of DNA from each sample was processed. The DNA was then amplified via a limited cycle PCR program, and AMPure XP beads (Beckman Coulter; Brea, CA, USA) were used for purification and size selection (>500 bp) of DNA fragments. To ensure the size selection was successful, 11 samples from the library were tested using the 2100 Bioanalyzer (Agilent Technologies; Santa Clara, CA, USA). The size-selected samples were then sequenced on the Illumina MiSeq platform (Illumina; CA, USA) using the MiSeq Reagent Kit v3. Sequencing reads were cleaned to remove those of low quality (minimum read length = 75, max ns = 1, minimum quality score = 17) and converted from fastQ to fastA using PrinSEQ [29]. Sequence data are available at NCBI's Sequence Read Archive using SRA identifier SRP216579 or BioProject number PRJNA556989. SRA and BioSample accession numbers for each sample can be found in Appendix A Table A1.

## 2.9. Functional Annotation of Metagenomes

Functional analysis of the metagenomes was performed using SUPER-FOCUS [30]. SUPER-FOCUS aligns sequence similarities using RAPSearch2 [31] and performs a 98% clustering of the proteins in the database to reduce the computation time. The output of SUPER-FOCUS provides functional assignments based on the SEED database. These hits are converted to the relative percent abundance for each site, and abundance data were the arc-sine square root transformed prior to statistical analysis [32].

## 2.10. Microbial Taxonomy Identification

Sequencing reads were aligned to the SEED database (http://theseed.org/wiki/Main_Page, accessed on 6 November 2020) using SUPER-FOCUS [24] for functional annotation, providing a SEED identifier containing an NCBI taxonomy ID (TaxonID) for each aligned read. Taxonkit (https://bioinf.shenwei.me/taxonkit/, accessed on 4 December 2020) was then used to extract TaxonIDs from SEED identifiers, retrieve relevant taxonomic information from NCBI and calculate abundance and relative abundance of each unique TaxonID for each metagenome (See Figure 2 under results ). Kingdom level classification was annotated using Kaiju in Greedy run mode, with a minimum match length of 11, a minimum match score of 90 and 5 allowed mismatches [33]. Microbial orders with abundances greater than 1% were utilized to increase the statistical power by reducing the possibility of Type 1 statistical errors [23].

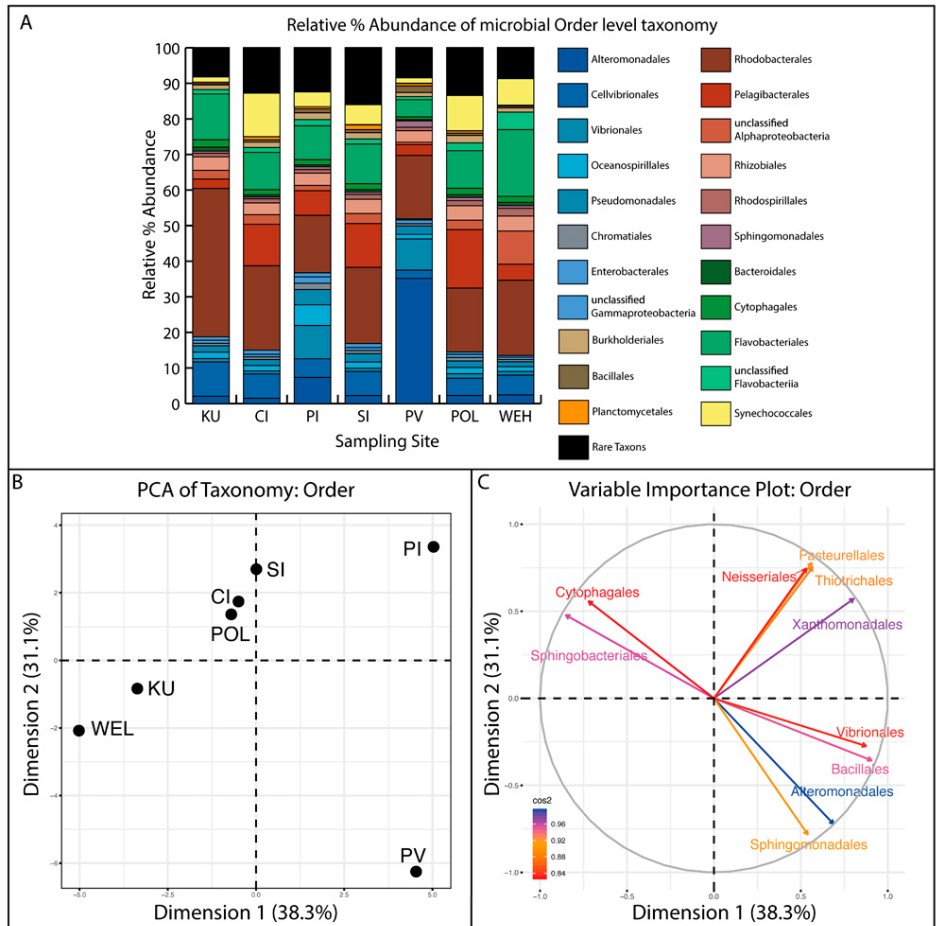

**Figure 2.** (**A**) Relative percent abundance of bacteria Order level taxonomy colored by Phylum. Blue hues represent Gammaproteobacteria, red hues represent Alphaproteobacteria and green hues represent Flavobacteria. (**B**) PCA of taxonomy at the Order level. (**C**) Variable importance plot of the top 10 orders.

*2.11. Statistics*

A Principal Component Analysis (PCA) was performed using relative abundance data to elucidate trends and groupings in both taxonomy and function using the R package bpca [34,35]. Variable importance plots were created to determine the taxa/functions driving the differences observed in the PCA using the R package FactoMineR [36]. Factors were ranked according to their $\cos^2$ value from 0 to 1, where factors with $\cos^2$ values closer to 1 were the most important drivers of PCA clustering. Linear regression models were used to determine correlation between benthic hard coral cover and normalized abundances of microbial metabolism using the base R "stats" package [37].

## 3. Results

*3.1. Reef Site Characteristics*

Seven coral reefs were sampled, four from the northeast end of Sri Lanka and three from the southern end (Figure 1A; Table 2). The percent live coral cover around Sri Lanka ranged from 16.5% at Paraviwella reef (PV) to 69.8% at Pigeon Island (PI); the mean across all sample sites (mean ± SD) was 43.8 ± 17.2%. The percent algae cover ranged from 6.4% at Small Island (SI) reef to 14.9% at Weligama reef (WEL) (Figure 1B, Table 2), with a mean across all sample sites (mean ± SD) of 10.3 ± 2.8%. Reef sites in the northeast KU, CI, PI and SI are in regions with less populated districts (61–200 persons per sq km). Weligama reef and Polhena reef (POL) in the south are in districts with high population densities (201–600 persons per sq km). The population densities were intermediate near the reef

site PV (601–100 persons per sq km). The PV reef is exposed to untreated sewage outflow and is in close proximity to a fishing harbor. The higher algal-dominated reefs (POL and WEL) were found closer to estuaries receiving catchment from the Nilwala and Polwaththe Rivers, respectively (Table 3).

**Table 3.** Summary of the temperature and salinity with the phosphate, nitrate, nitrite and ammonia concentrations measured for the coral reef sites in Sri Lanka during the study period.

| Coral Reef Site. | Sample Month | Temp/°C Range | Salinity (ppt) | Phosphate µM | Nitrite + Nitrate µM | Ammonia µM |
|---|---|---|---|---|---|---|
| Kutchchaweli (KU) | | | 28 | $0.24 \pm 0.02$ | $1.76 \pm 0.37$ | $0.72 \pm 0.19$ |
| Coral Island (CI) | | | 28 | $0.06 \pm 0.006$ | $0.21 \pm 0.06$ | $0.34 \pm 0.44$ |
| Pigeon Island (PI) Trincomalee | June 2015 | 30–31.5 | 28 | $0.14 \pm 0.02$ | $2.14 \pm 0.09$ | $1.08 \pm 0.66$ |
| Small Island (SI) | | | 28 | $0.08 \pm 0.008$ | $0.37 \pm 0.16$ | $0.20 \pm 0.17$ |
| Paraviwella (PV) | February 2016 | | 32 | $0.61 \pm 0.01$ | $9.78 \pm 0.18$ | $4.29 \pm 2.00$ |
| Polhena (POL) | February 2016 | 29.5–31 | 31 | $0.19 \pm 0.005$ | $2.39 \pm 0.08$ | $1.56 \pm 0.99$ |
| Weligama (WEL) | March 2016 | | 28 | $0.15 \pm 0.006$ | $1.58 \pm 0.23$ | $1.24 \pm 1.20$ |

Sites in the northeast tended to have higher coral cover (ranging between 49.7 and 69.75%, mean $\pm$ SD 55.6 $\pm$ 9.5%) than sites in the south (ranging between 16.4 and 36.2%, mean $\pm$ SD 28.05 $\pm$ 10.3%). The algal cover was lower in the northeast (ranging from 6.4 to 10.4%, mean $\pm$ SD 8.5 $\pm$ 1.6%) than in the south (ranging from 10.1 to 14.9%, mean $\pm$ SD 12.6 $\pm$ 2.4%).

### 3.1.1. Reef Water Chemistry

The temperature in the north ranged between 30 and 31.5 °C and in the south from 29.5 to 31 °C (Table 2). Water was collected to determine the levels of ammonia, nitrate + nitrite (NOx) and phosphate present at each reef (Figure 1C, Table 3). Overall, the nutrient levels did not differ much between reefs, with the exception of Paraviwella (PV), a shallow granite dominant reef in close proximity to domestic sewage run off, which had comparatively high levels of ammonia and $NO_x$, at 4.29($\pm$2) µM and 9.78($\pm$0.18) µM, respectively. However this difference was not statistically significant. The phosphate concentrations ranged between 0.06($\pm$0.006) µM to 0.61($\pm$0.01) µM, with the highest phosphate concentration being from the Paraviwella (PV) reef. When sites from the north were compared to sites from the south to detect any differences in the chemistry of water due to the location as several river run off vicinity to reefs, no significant differences were detected for any nutrient ($n = 7$; Welch's two sample *t*-test; ammonia: $p = 0.22$, df = 2.11; NOx: $p = 0.31$, df = 2.14; Phosphate: $p = 0.33$, df = 2.29). The complete nutrient measurements are found in Table 3.

### 3.1.2. Microbial Community Structure and Function

Seawater samples collected from 1 m above the benthos at each reef site produced seven metagenomic libraries that comprised 4.17 million quality reads ranging from 340,000 to 1.4 million reads per reef. These reads were used for taxonomic and metabolic profiling of the Microbial communities present in the water column (Table 3).

Epifluorescence microscopy direct counts showing that highest bacterial particle counts were reported from KU reef and the lowest reported from PV reef ($8.3 \times 10^5$ mL and $1.4 \times 10^5$ mL respectively) (Table 4). No counts were performed for Viruses, Euckaryots and Archea.

**Table 4.** Microbial abundance ($\times 10^5$) and %GC content per site.

| Site | Microbe ($\times 10^5$) per mL | % GC Content |
|---|---|---|
| KU | 8.3 | 47.0 |
| CI | 4.5 | 44.0 |
| PI | 2.9 | 46.6 |
| SI | 5.9 | 44.0 |
| PV | 1.4 | 47.9 |
| POL | 2.2 | 44.3 |
| WEL | 7.1 | 46.5 |

Taxonomy was assigned from the sequence data using Kaiju. Between 46.1% (POL) and 71.8% (PV) of reads were assigned to a domain (average of 56.8%). Of the assigned reads, most reads were assigned to microbial (95.4%), followed by Eukaryotes (1.9% of assigned reads), Archaea (1.6% of assigned reads) and viruses (0.8% of assigned reads) (Tables 4 and 5). Taxonomic assignments of the microbial community were analyzed at various levels, as described in the Methods. Figure 2A illustrates the bacteria community structure at the order level. Alphaproteobacteria (34.2 ± 3.2%) and Gammaproteobacteria (23.9 ± 5.6%) were the most abundant classes at all sites, accounting in combination for up to 79.4% of all taxa present. At the order level, Rhodobacterales dominated the microbial community across all sites, with the exception of the Paraviwella reef, in which the order Alteromonadales was most abundant.

**Table 5.** Summary of taxonomic assignments to kingdoms from Kaiju. Kuchchaweli (KU), Coral Island (CI), Pigeon Island (PI) and Small Island (SI) belongs to Trincomalee District. Weligama reef (WEL) and Polhena reef (POL) in the south belong to Matara District. Paraviwella reef site (PV) in the south belongs to Hamabnatota District.

| Site | Post-QC Reads | Assigned | Unassigned | Bacteria | Eukaryota | Archaea | Viruses |
|---|---|---|---|---|---|---|---|
| KU | 806,743 | 497,202 | 309,541 | 487,156 | 3692 | 4491 | 1234 |
| CI | 373,567 | 212,173 | 161,394 | 199,761 | 5967 | 2377 | 3670 |
| PI | 361,566 | 193,905 | 167,661 | 188,378 | 2387 | 1016 | 1756 |
| SI | 623,758 | 294,601 | 329,157 | 275,545 | 8739 | 7727 | 1968 |
| PV | 326,126 | 234,300 | 91,826 | 230,327 | 1167 | 1441 | 1212 |
| POL | 1,058,343 | 487,603 | 570,740 | 443,761 | 19,877 | 15,254 | 7198 |
| WEL | 693,647 | 423,146 | 270,501 | 406,118 | 5261 | 10,069 | 869 |

A Principal Coordinate Analysis (PCA) and Variable Importance plot of order-level taxonomy was performed and identified the 10 most influential taxonomic orders across all sites (arrows in Figure 2C). Sites POL, CI and SI clustered separately from sites KU and WEL (Figure 3B). Paraviwella reef (PV) and Pigeon Island reef (PI) were the most disparate along both component 1 and 2 and clustered independently of all other sites. Figure 3B depicts the top 10 most important orders as they relate to the coral reefs in ordination space. Order Alteromonadales was the most important driver of reef clustering, illustrated by its $\cos^2$ value of nearly 1. The orders Sphingobacterales and Cytophagales were negatively correlated with orders Alteromonadales and Vibrionales and were the most important factors for clustering of five out of the seven reefs sampled.

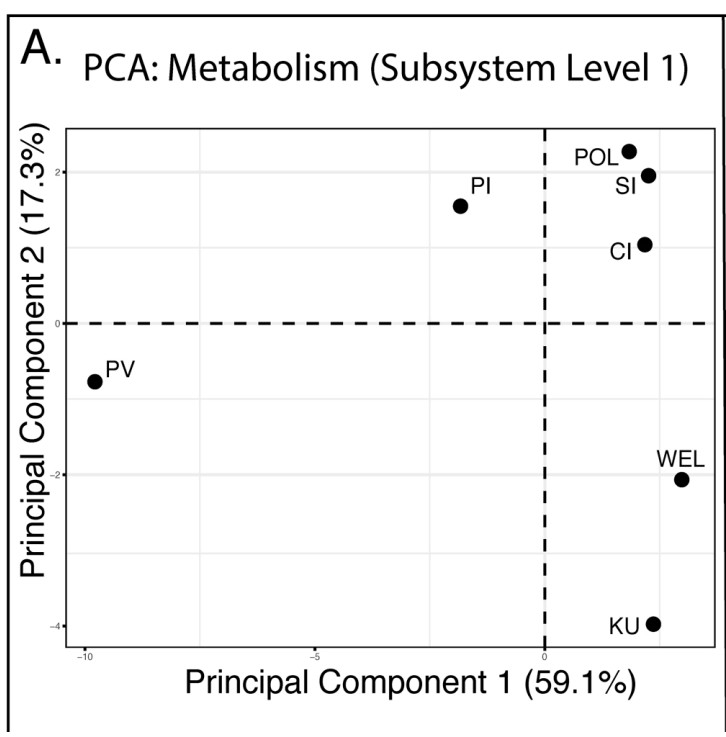

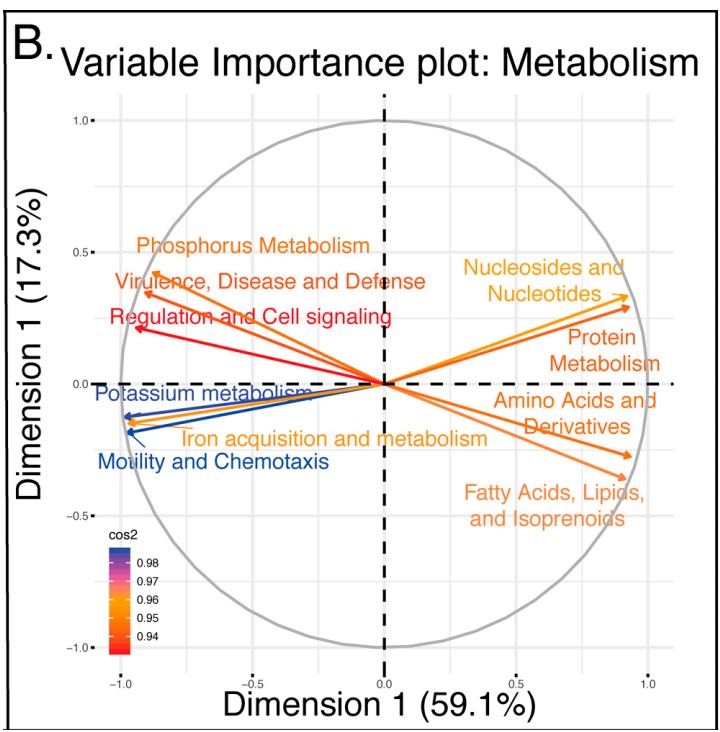

**Figure 3.** (**A**) PCA of SEED Subsystem Level 1 genes across all sites. (**B**) SEED Subsystem Level 1 Variable importance plot.

A Principal Coordinate Analysis (PCA) and Variable importance plot of function were performed (Figure 3A,B). Function data were based on SEED subsystem level 1, which has 30 categories and 1358 annotated subsystems. Each reef site had reads corresponding to all metabolic categories, except for "Plant cell wall and outer surfaces". In the resultant PCA plot, samples from coral reefs at Polhena (POL), Small Island (SI), Pigeon Island (PI) and Coral Island (CI) clustered together near the upper right quadrant; this grouping was mainly driven by Protein Metabolism and Nucleosides & Nucleotides. The cluster

of Weligama reef (WEL) and Kuchchaweli reef (KU) in the lower right quadrant was significant based on SIMPROF (Figure 3) and was driven by the metabolism of aromatic compounds, lipid metabolism and amino acid metabolism. Most dissimilar to any other reef was Paraviwella reef, whose position at the far left was due to this reef possessing higher relative abundances of functions such as motility and chemotaxis and the stress response pathways.

## 4. Discussion

### 4.1. Coral Reefs Status in South Asian Region

The South Asian region accounts for 4.2% (10,949 km$^2$) of the global area of coral reefs. These reefs are distributed in Bangladesh, India, Maldives, Myanmar, Pakistan and Sri Lanka and the Chagos Archipelago in the Indian Ocean. Much of the reef area is concentrated along the Lakshadweep–Maldives–Chagos Ridge, which accounts for around 75% of the total reef area in the region [38]. Other significant reef systems are found in the Gulf of Mannar and around parts of Sri Lanka. Reef development is poor along mainland India, Pakistan and Bangladesh.

Coral reef research is in the developing stage due to the issues related to capacity and infrastructure under developing economies. Reef monitoring was initiated to detect the major disturbances such as global bleaching events and Indian Ocean Tsunami 2004. Systematic monitoring commenced in 1998 following bleaching events in South Asia including Sri Lanka [13,19,38].

The South Asian regional coral cover is in declining trend although reefs showed recovery after bleaching according to the Global Coral Reef Monitoring status report 2022. The average cover of algae progressively increased since 2015, and there were upward trends in algal cover corresponding with a decline of live coral cover.

### 4.2. Population Densities of South Asian Region

In addition, coastal fringing reefs along Sri Lanka including India, Bangladesh and Pakistan, to suffer from anthropogenic stresses compared to the offshore island groups along the Lakshadweep-Maldives-Chagos ridge have less anthropogenic pressure; many of them restricted access or declared as marine parks. South Asia is characterized by a high human population and high human population densities. The total population of the region exceeds 1.8 billion, with densities ranging from 244 people per km$^2$ in Pakistan to more than 1100 people per km$^2$ in Maldives and Bangladesh. Sri Lankan human population density is 341 km$^2$, this value is relatively high for a larger island [38].

### 4.3. Sri Lankan Reefs and Long Term Impacts

Sri Lankan reefs are in a unique geographical setting in the Indian Ocean, where reefs have shown little or no recovery after a bleaching event in 1998 due to chronic stress from natural and anthropogenic stress. Prevalence to natural and anthropogenic impacts to reefs in Sri Lanka continuously not allowing time for recovery as one disturbance occurs before experiencing another. The microbial communities associated with the water column of the reef locations in Sri Lanka subjected to gradients of natural and anthropogenic impacts could lead to development of understanding on the status and functions of coral reefs.

### 4.4. Microbial Taxonomic Community Composition Structure of Sri Lanka Northeast and South Regions of Sri Lanka

Early studies of Sri Lankan reefs examined the taxonomic diversity, status of corals [3,8,9,12] and algae cover with respect to monitoring [12]. Several researchers have identified major and minor human impacts leading to reef degradation including pollution, unsustainable fishing with respect to human population densities at urban and rural cities of the coastal area [9,15]. Coral bleaching and recovery from exposure to tsunami in 2004 are notable incidents affecting the coral health status [10,14]. The behavior of microbial communities in the water column in healthy versus degraded reefs were compared with the other oceanic reefs

and noted similarities to fit with the proposed DDAM model on global microbialization [19]. However, the taxonomic composition of the microbial communities was not studied for these reefs across Sri Lanka associated with a range of human population densities along the coastline.

Numerous studies on microbes on coral reefs have shown that microbial community structure differs between intact and degraded reefs [21–24]. During this study the first data set for the microbial community taxonomy is presented for Sri Lankan reefs.

Taxonomic analysis showed that reefs SI, CI, POL, KU, and WEL were associated with orders Sphingobacteriales and Cytophagales. Sphingobacteriales are important for the marine nitrogen cycle, as they fix nitrogen in nitrogen limited systems. The order Cytophagales possess the ability to process complex biomolecules and they play a critical role in organic matter turnover. The microbes on these reefs may getting biomolecules from coral, algae and other complex sources. PI and PV reefs did not possess abundant taxa in nitrogen fixation or biomolecule turnover and these reefs need to be further studied to establish correlation with microbial abundance with taxonomic data to understand the impact microbial taxonomy plays on overall reef health and energy flow in the system.

However, Paraviwella (PV) reef had abundant Vibrionales and Alteromonadales, which include copiotrophic members, representing 30% of the total microbial community (Figure 2A). These copiotrophic microbes possess large genomes and are often resistant to heavy metals, however this reef is not reported with such heavy metal pollution. These traits may be advantageous at this reef site due to its proximity to the outflows of untreated domestic sewage and as a result, allowing this microbe to thrive in an environment with increased toxicity and nutrients.

*4.5. Need of a Comprehensive Study to Understand the Energy Utilization by Microbial System in Relation with Benthic Coral Cover to Eliminate Limitations of This First Baseline Data Set*

These results from the present study may be confounded by particular variables, such as DOC level [39–41], temperature, salinity, and turbidity. These data were available for some sites but were ultimately excluded from the analysis because of low sample size, and thus low statistical power. Increased sampling around Sri Lanka, at additional sites and multiple samples taken from each site are needed to provide enough data to address the potential effects of these environmental variables on the microbial communities [42]. Thus, because of low sample size, our dataset lacked the power to detect statistically significant differences across regions (northeast vs. south). Correlational analyses of all sites were completed in lieu of examining regional differences including other variables (e.g., temperature, salinity). The benthic monitoring data used in this study is similar to the present monitoring data. The reefs in the northeast represent higher coral cover and lower algae cover compared to the reefs in the south dominated with macroalgae. Major river discharges to ocean via estuaries continue to supply materials leads to reef degradation. The human population densities published were not updated after 2012, there may be some increment can be expected as per the statistical hand book published by the department of census and statistics indicated that the human population density as 353 persons per square kilometer. The results from this study should serve as the basis for future investigations of these understudied but interesting tropical reef systems in the south Asian region in the Indian Ocean supporting blue economies [43].

## 5. Conclusions

This study presented microbial and benthic data from seven Sri Lankan coral reefs an Indian Ocean island with high population densities exposed to varying natural and anthropogenic impacts over several decades. The results from this study shows the dominant order level community structure of the reef microbiome for the first time, however the bacterial community taxonomy and metabolism are not intrinsically linked. Continual monitoring and data collection on viruses, archaea, need to be pursued in order to fully understand the fluctuating microbial communities present on these reefs and their

functional role and bioenergetics. Due to the near-shore locations of the reefs around Sri Lanka, sample collection from various seasons and time points throughout the year can be logistically less challenging than remote atoll sample collection, but current political and cultural pressures of the region may hinder this endeavor. This unique reef system around Sri Lanka presents an opportunity to better understand the microbiology of coral reefs with long histories of human impact and inhabitation, and to understand how reefs cope with increased stressors. In terms of conservation and management of reefs in Sri Lankan reefs, we recommend civil engineering manipulations to control river surface runoff and minimize entering land based pollutants to reef environments.

**Author Contributions:** M.F.M.F. and K.T.G. conceptualized the study, analyzed data and wrote the paper. S.M., D.S.N., A.C.-G., K.N.M.S., A.M.K.N.K., W.A.S.C. and L.W.-K. performed experiments and analyzed data. F.R. contributed to the concept and design of the study, data analysis and manuscript writing. All authors have read and agreed to the published version of the manuscript.

**Funding:** This research was funded by a Gordon Betty Moor Foundation under the Marine Microbiology Initiative Investigator award (GBMF#3781), Aquatic Symbiosis Investigator Award (GBMF9207) and exploring the origin of viruses (GBMF9871) to FLR.

**Institutional Review Board Statement:** Not applicable.

**Informed Consent Statement:** Not applicable.

**Data Availability Statement:** Sequence data are available at NCBI's Sequence Read Archive using SRA identifier SRP216579 or BioProject number PRJNA556989. SRA and BioSample accession numbers for each sample can be found in Appendix A.

**Acknowledgments:** Boat operators and Fisheries societies in Trincomalee and Kalpitiya/Puttalam Sri Lanka are hereby acknowledged for their support in the field work.

**Conflicts of Interest:** The funders had no role in the design of the study; in the collection, analyses or interpretation of the data; in the writing of the manuscript; or in the decision to publish the results.

## Appendix A

**Table A1.** Details of NCBI's Sequence data from this study with Sequence Read Archive (SRA) numbers and BioSample numbers.

| Site | SRA Number | BioSample Numbr |
|------|-----------|------------------|
| CI | SRS169408 | SAMN12385965 |
| KU | SRS169407 | SAMN12385964 |
| PI | SRS169405 | SAMN12385966 |
| POL | SRS169404 | SAMN12385969 |
| PV | SRS169403 | SAMN12385968 |
| SI | SRS169406 | SAMN12385967 |
| WEL | SRS169402 | SAMN12385970 |

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
