# Peer review of "Microbiome of Sri Lankan Coral Reefs: An Indian Ocean Island Subjected to a Gradient of Natural and Anthropogenic Impacts"

_2673-1924, doi:10.3390/oceans4020013_

Round 1

Reviewer 1 Report

Summary and general comments:

This article presents bacterial taxonomic information from metagenomic libraries created from coral reef water samples from Sri Lanka. The authors also present environmental data from the same sites, including nutrient concentrations, benthic cover and human population size. This is an exciting dataset and will be important for environmental managers in Sri Lanka. However, currently the dataset is not used to its full potential. The authors only present taxonomic data, and do not make use of the metagenomic data they have to investigate functional differences in reef microbial communities. Further, they do not statistically link the patterns in the environmental data with the patterns in microbial community structure. I have provided some possible ways to do this below. With a little extra work this manuscript could be a great addition to the coral reef literature from Sri Lanka.

Specific comments:

Line 1: The title is misleading, it makes it sound like you are investigating the microbiome in relation to the anthropogenic gradient, but this is not fully drawn out in the analyses. Please change to represent the work done or do some further analyses.

Line 14: You do not present any results on metabolic makeup, you only discuss taxonomy. Please remove reference to functional analysis throughout the manuscript or do the analysis.

Line 28: This reads better as two sentences, with a full stop after sub-continent.

Line 29-32 and 35-37: What is the relevance of telling the reader about the history of these reefs if it doesn't come up again? Please remove to streamline the text.

Line 27-81: The introduction is a little disjointed at present. You discusses human impacts in multiple paragraphs - it would be better to pull this together in one or two concise paragraphs and streamline.

Line 67: You say that there are no previous attempts to characterise the microbial communities of Sri Lanka coral reefs but then in the discussion you reference the Hass paper (Line 265), which does exactly that.

Line 98: You mention direct counts of Bacteria and Archaea but then never discusses this again. Take out if not relevant to this paper.

Line 99: Please add a reference for Hatay Niskin bottle.

Line 174-180: Take out if you do not discuss functional findings in this paper.

Line 204: Statistics for population sizes not presented. Please add.

Line 207: where not were

Line 208: Again, statistics for coral and algal cover?

Line 224: Take out 'function' as you do not discuss this.

Line 233: Please use 'bacterial' instead of 'microbial' throughout if you are only presenting bacterial data.

Line 196-258: There is currently a clear lack of analysis that compares the taxonomic data to the environmental variables that you measured. Please provide an explanation for this or add some extra analyses. This paper would be greatly strengthened by an analysis that attempts to elucidate the reasons (e.g. nutrients, human populations) for differences in community structure. To do this the authors could explore a distance-based redundancy ordination analysis (dbRDA) using Bray-Curtis distances with the capscale function in the ‘vegan’ package v2.5.7 (Oksanen et al., 2013). You could also use the multipatt function in the R package "indicspecies" v1.7.9.

Line 257: At no point do you discuss the different sampling dates of the reefs. Could this play a significant role in the different community you find at PV? Please discuss.

Line 258: You present eukaryotic, and viral data here but make no reference to it in the text. Please remove or discuss in more detail.

Line 262: “remained relatively similar in response” – how do you know this is in response to benthic composition? You have not presented any statistics to show that this is the case. Please reword.

Line 265: You start this paragraph talking about degraded vs intact reefs, but then do not link this to your own sites and instead switch to talking about certain sites and orders of bacteria. The link is not clear in this paragraph, please reword.

Line 268 onwards: You have the data to back up your discussion points here, so why not use it?

Line 272: You contradict your argument about heavy metal pollution in this paragraph – first you imply the microbes are there due to resistance to heavy metals and then they say that heavy metals are not found – could there be another explaination? Could you use metagenomic data to back this statement up?

Line 279-282: The results make it difficult to draw conclusions about which reefs are degraded and which are not. Can you please add some statistics in here to show that some reefs have a significantly more degraded profile than other reefs? And again how are you accounting for difference in sampling month? This statement is not supported without more clarity throughout.

Line 284-291: Please link to the work presented here.

Line 293: This does not follow on from what is being discussed above.

Author Response

Thank you for the very constructive comments.

Responses are attached for your perusal please

Fairoz

Reviewer 2 Report

General comments

The submitted manuscript entitled “Microbiome of Sri Lankan Coral Reefs: An Indian Ocean Island subjected to a gradient of natural and anthropogenic impacts” deals with the investigation of reef benthic structure and microbial communities in the water column using metagenomics. The authors have monitored Sri Lanka’s coastal coral reefs by sampling the water column at seven reef sites on the northeastern and southern coasts of Sri Lanka. The authors analyzed the microbial metagenomes to elucidate both the taxonomic and metabolic makeup of the microbial communities present at each site.

The general manuscript is well-written and provides interesting results. However, the authors must pay attention to the journal format and minor grammatical errors throughout the manuscript. I have a few concerns, and I have elaborated on them here.

Major comments:

The authors must improve the introduction section. The authors must elaborate on previous works on the reef system, particularly in the northeastern and southern coasts of Sri Lanka, in context to their manuscript.

For instance, the authors can specifically elaborate on previous works, as mentioned in Lines 67-71: “However, there are no previous studies that have identified or characterized the microbial communities of coral reefs in Sri Lanka. There may have also been substantial changes to the reef system of Sri Lanka since the previous studies of reefs were conducted, as major natural impacts such as a coral bleaching event in 1998 and a tsunami in 2004 [10] have occurred.”

The figures and tables must be formatted appropriately in the manuscript. For instance, the figures and tables must be close to where its first mentioned in the text and not in the results section.

The authors must improve their writing style in the results section, particularly section 3.1.1.

Lines 215-217: “Overall, nutrient levels did not differ much between reefs with the exception of Paraviwella (PV), which had comparatively high levels of ammonia and NOx, at 4.29(±2) μM and 9.78(±0.18) μM,”: reason?

Lines 220-222: “When sites from the north were compared to sites from the south, no significant differences were detected for any nutrient”: The authors must reflect on the significance of studying the nutrient content and attempt to explain their results.

Lines 237-239: At the order level, Rhodobacterales dominated the microbial community across all sites, with the exception of the Paraviwella reef, in which the order Alteromonadales was most abundant.: reason?

The authors must elaborate on the natural and anthropogenic impacts in their discussion and conclusions sections.

Minor comments:

Line 9: “from the Indian Ocean continental island, Sri Lanka” may be referred to here when Sri Lanka is first mentioned.

Lines 35-37: “Human dependence on Sri Lankan reefs may have a long history, as archaeological evidence suggests modern Homo sapiens inhabited Sri Lanka 8,000 years ago [7,8,9].” The authors must reconsider if this information supports their manuscript and if the age reference to 8,000 years ago is correct.

Line 43: “Indian Ocean [10]).” Formatting error.

Lines 59-63: Consider rewriting the long sentence for clarity.

Figure 1A is missing latitude and longitude coordinates.

Lines 87-88: “All reef sites in this study were previously included in monitoring studies [19, 20, 12, and 14]. Water sample collection and processing.”: Please consider the formatting.

Lines 107, 149, 152, 213: for example, “at -20o C.” and “into a 55 oC”: Please consider the formatting throughout the manuscript.

Line 198: “four from the northern end of Sri Lanka”: The authors must consistently use northeast throughout the manuscript.

Line 205: “south reef sites were similar (61-200 and 201-600 persons per sq km, respectively”: The range 61-200 and 201-600 persons per sq km are not similar.

Line 206: “(POL and WEL)”: Abbreviations are not defined.

Line 224: “3.1.1. Microbial community structure and function”: It should be 3.1.2.

Author Response

Dear Reviewer 2,

Thank you for your constructive comments and reply/ responses are attched for your perusal please.

Fairoz

Reviewer 3 Report

Authors make a good argument for the relationship between coral reef health with the presence of human populations. There were some obvious differences in coral cover (high) and ammonia (low) between northern sites (less populated) and southern sites (more populated). There were no correlates with microbial abundance or types in this study. 

Interesting results: Three northern and two southern reefs had Sphingobacterales that fix nitrogen and Cytophagales that play a critical role in organic matter turnover. One northern and one southern reef did not possess abundant taxa in nitrogen fixation or biomolecule turnover. One southern reef also had abundant Vibrionales and Alteromonadales, which probably indicated the site’s proximity to the outflows of untreated domestic sewage.

The microbial systems of northern and southern sites did not correlate well with the health of degraded or healthy reefs in this study. Disappointing but not unexpected.

Sri Lanka’s reefs are in close proximity to high population densities. I was looking for recommendations to use to sustainably manage and initiate conservation priorities. Do the authors have any?

I also note that the data was taken 7 years ago, and the population data is 10 years old. A lot can happen in that time! Is the study still relevant?

Author Response

Dear Reviewer 3,

Thank you for your valuable comments to improve the manuscript. We appreciate your recognition on the results on the microbial taxa and its functional role in the reefs studied. We agree that we cannot make any correlation with the microbial abundance with available data and in future we will expand the data set for such analysis.

We have included the data from the past years, however the present monitoring shows that the same status of reef condition considering the coral and algae cover. Human population data were not been updated since 2012 and the numbers may have increased by now with reference to the brief reporting through the “Statistical hand book of Sri Lanka” published in 2021 by the Registrar General Department. It further state that the human population density as 353 per square kilometer.

We have also included on the minimization of coastal pollutants enter to reef areas by civil engineering manipulations by constructing catchment basins/ river basin management and that would be a costly task for developing economies. Also more sewage treatment facilities need to be implemented as a strategy to minimize coastal pollutants enters to oceans.  

Sincerely

MFM Fairoz   

Round 2

Reviewer 2 Report

The resubmitted manuscript titled “Microbiome of Sri Lankan Coral Reefs: An Indian Ocean Island subjected to a gradient of natural and anthropogenic impacts” was previously recommended for a minor revision.

Following, I have included the comments.

Major Comments:

The resubmitted manuscript does not show significant improvement. This is because the editing done by the authors needs extensive editing in the English language and style such that their intended progress is difficult to understand and is mostly incomprehensible. This is evident in lines 37-38, 42-44, 60-64, 66-69, 72-74, 231-233, 327-334, 356-357, and 365-372. Therefore, I have significant concerns over the English language and style in the manuscript.

The authors have not significantly improved the introduction section by elaborating on previous works on the reef system. Also, I do not understand the significance of using the “unpublished” phrase in the manuscript (Lines 68, 231, and 366). Moreover, the discussion and conclusions sections also fail to elaborate on the natural and anthropogenic impacts.

Minor Comment:

Line 19: “Principal coordinate analysis (PCA)..”: PCA here refers to Principal coordinate analysis or Principal Components Analysis?

Author Response

Dear Reviewer,

On behalf of all co-authors I would like to thank you for your constructive comments and that helped to improve the manuscript to the present status. We wish all the comments and suggestions were incorporated including corrections on English language.

Thank you

Sincerely

Fairoz

Round 3

Reviewer 2 Report

.

Author Response

Dear Reviewer,

Thank you for your constructive comments, We have done changes accordingly in order to :

  1. Correct language
  2. Improve the results sections to support with conclusions and how the study could recommend for conservation and management of coral reefs in Sri Lanka.
  3. This study is the first data set and it could lead to more details with the ongoing monitoring in Sri Lanka 

Sincerely

MFM Fairoz